# A Novel Heat Shock Transcription Factor (*ZmHsf08*) Negatively Regulates Salt and Drought Stress Responses in Maize

**DOI:** 10.3390/ijms222111922

**Published:** 2021-11-03

**Authors:** Jing Wang, Li Chen, Yun Long, Weina Si, Beijiu Cheng, Haiyang Jiang

**Affiliations:** National Engineering Laboratory of Crop Stress Resistance Breeding, School of Life Sciences, Anhui Agricultural University, Hefei 230036, China; wangjing@stu.ahau.edu.cn (J.W.); cl0226@stu.ahau.edu.cn (L.C.); 20720531@stu.ahau.edu.cn (Y.L.); weinasi@ahau.edu.cn (W.S.)

**Keywords:** maize, *ZmHsf08*, transcription factor, salt stress, drought stress

## Abstract

Heat shock transcription factors (HSFs) play important roles in plant growth, development, and stress responses. However, the function of these transcription factors in abiotic stress responses in maize (*Zea mays*) remains largely unknown. In this study, we characterized a novel HSF transcription factor gene, *ZmHsf08*, from maize. *ZmHsf08* was highly homologous to SbHsfB1, BdHsfB1, and OsHsfB1, and has no transcriptional activation activity. The expression profiles demonstrated that *ZmHsf08* was differentially expressed in various organs of maize and was induced by salt, drought, and abscisic acid (ABA) treatment. Moreover, the overexpression of *ZmHsf08* in maize resulted in enhanced sensitivity to salt and drought stresses, displaying lower survival rates, higher reactive oxygen species (ROS) levels, and increased malondialdehyde (MDA) contents compared with wild-type (WT) plants. Furthermore, RT-qPCR analyses revealed that *ZmHsf08* negatively regulates a number of stress/ABA-responsive genes under salt and drought stress conditions. Collectively, these results indicate that *ZmHsf08* plays a negative role in response to salt and drought stresses in maize.

## 1. Introduction

Plants are prone to encountering environmental stresses during all aspects of plant growth and development, resulting in devastating damage to the survival and production of the plants. Among these environmental factors, drought and salt stress are the most important challenges limiting crop growth and grain yield [1,2]. On the one hand, salt and drought stress cause dehydration, which in its turn results in ionic and osmotic stress, destroying cellular homeostasis; on the other hand, salt and drought stress trigger oxidative stress, which produces reactive oxygen species (ROS) and provokes damages to membrane lipids, proteins, and DNA, resulting in plant wilting and even plant death [3,4].

Plants have developed sophisticated tolerance mechanisms to survive drought and salt stress. Many key factors are involved in drought and salt stress responses, including kinases [5,6], ion transporters [7,8], ROS [3,9], abscisic acid (ABA) [10,11], and transcription factors (TFs) [12,13]. A wealth of studies has demonstrated that many TF families, such as dehydration-responsive element-binding protein (DREB), WRKY, v-myb avian myeloblastosis viral oncogene homolog (MYB), basic leucine zipper (bZIP), and HSF, play a pivotal role in the transcriptional regulation of genes during abiotic stress responses [14,15,16,17,18]. TFs activate the transcription of stress-responsive genes by binding to their promoters, resulting in enhanced stress tolerance in plants [19]. HSFs are an important gene family of TFs, which are involved in plant growth, development, and stress response. Since the first plant HSF was isolated from tomato in the 1990s, a large number of plant HSFs have been identified and cloned [20]. For example, there are 21 HSFs in Arabidopsis that have been identified [21], 38 in soybean [22], 31 in maize [23], and 25 in rice [24]. Moreover, the maximum number of HSF families in plants was identified in wheat: at least 56 HSFs [25].

Generally, plant HSFs have conserved functional domains, including the N-terminal DNA binding domain (DBD), oligomerization domain (OD or HR-A/B), nuclear localization signal (NLS), and nuclear export signal (NES) [21,26]. In addition, plant HSFs are classified into three classes (HSFA, HSFB, and HSFC) based on the sequence characteristics of the OD domain. Interestingly, the HSFAs have short activator motifs (AHA motifs) located in their C-terminal domains, and they function as transcription activators, while class B and C HSFs lack the activation domain [18,21]. For the past few years, many studies reported the critical roles of HSFAs in plant response to various abiotic stresses. It has been determined in tomato and Arabidopsis that *HsfA1s* function as the master regulators in plant heat stress response (HSR) and are prerequisite for the activation of transcriptional networks [27,28,29]. The overexpression of *AtHsfA1b* enhanced the water productivity of transgenic plants, resulting in increased resistance to drought [30]. *HsfA2s*, the direct target genes of *HsfA1s*, were reported as key regulators in plant HSR and other abiotic stress responses [18,31]. Overexpression of *HsfA2* enhanced thermotolerance of transgenic plants [32,33,34,35]. Ectopic expression of rice *OsHsfA2e* in Arabidopsis enhanced thermotolerance and salt stress tolerance of transgenic plants [36]. Furthermore, *HsfA3*, *HsfA6b*, *HsfA6f*, and *HsfA7b* also play an important role in abiotic stress tolerance in Arabidopsis, wheat, and rice [37,38,39,40,41,42].

In contrast to HSFAs, there was no evidence that HSFBs functioned as transcription activators on their own. In tomato, HsfB1 has been reported to act as a coactivator cooperating with HsfA1a or forming a functional triad with HsfA1a and HsfA2, regulating the expression of heat stress (HS)-responsive genes during plant HS responses [21,43,44]. Moreover, HsfB1 also cooperates with other transcriptional regulators to control target gene expression in abiotic stress responses. For example, ectopic expression of chickpea *HSFB2* in Arabidopsis improved the drought tolerance of transgenic plants by increasing the expression of stress-responsive genes such as *RD22*, *RD26*, and *RD29A* [45]. Decreased drought and salt tolerances were observed in rice plants over-expressing *OsHsfB2b* [46]. A study of HSF interaction showed that HsfB1 and HsfB2b have the capability to form complexes in vivo and might be involved in complex regulatory networks of signaling processes and stress responses [47].

Recently, 31 ZmHSFs were identified in maize and divided into 16 subclass A, 10 subclass B, and 5 subclass C gene members [48]. Ectopic expression of maize *ZmHsf05* (members of subclass A2) in Arabidopsis improve the heat tolerance of transgenic plants [49,50]. Furthermore, overexpression of a member of subclass A1 (*ZmHsf06*) in Arabidopsis enhances its salt stress tolerance [51]. However, the roles of ZmHSFs in plants’ salt and drought tolerance are largely unknown. In this study, we cloned *ZmHsf08*, a member of subclass B1, and analyzed its function in salt and drought stress responses. We found that *ZmHsf08* was induced by NaCl and PEG treatments and localized to the nucleus. Notably, overexpression of *ZmHsf08* in maize significantly enhanced sensitivity to both salt and drought stresses. Our results suggest that *ZmHSF08* plays a negative role in regulation of salt and drought responses and serves as a gene resource for the improvement of maize against abiotic stresses in the future.

## 2. Results

### 2.1. Gene Isolation and Sequence Analysis of ZmHsf08

The coding sequence of *ZmHsf08* was cloned from seedlings of the maize inbred B73. Sanger sequencing results showed that the coding sequence (CDS) of *ZmHsf08* was 897 bp in length (Figure 1A). *ZmHsf08* harbored 298 amino acids, including a conserved N-terminal DNA binding domain (DBD), an oligomerization domain (OD), a nuclear export signal (NES), and a nuclear location signal (NLS) (Figure 1B). Further multiple sequence alignments and homologous analysis revealed that *ZmHsf08* exhibited high sequence similarity with other known HsfB1 proteins, homologous to the B1 subclass of ZmHsfs (Figure 1B,C). Most importantly, *ZmHsf08* contained the core B3 repression domain (BRD, K/SV/KLFGVL/LTD), which is conserved in HsfB1 orthologs of five plant species (Figure 1B).

### 2.2. Subcellular Localization of ZmHsf08

To further analyze the subcellular localization of *ZmHsf08*, we constructed fusion proteins containing C-terminal and N-terminal green fluorescent protein (GFP), respectively (Figure 2A). The recombinant protein was transiently expressed in maize protoplasts. The p1305-GFP and PMDC43, used as the control vectors, were expressed in the nucleus and cytoplasm. Both *ZmHsf08*-GFP and GFP-*ZmHsf08* fluorescence signals were only concentrated in the nucleus, indicating that *ZmHsf08* was a nucleus-localized protein (Figure 2B).

### 2.3. Tissue-Specific and Stress-Induced Expression Profiles of ZmHsf08

To investigate the tissue-specific expression of *ZmHsf08* in maize, we investigated the expression profiles of *ZmHsf08* in six tissues (roots, stem, leaves, corn-silk, tassels, and bract) by RT-qPCR (Figure 3A). The results indicated that the *ZmHsf08* constitutively expressed in all surveyed six tissues under the non-stress condition.

To explore the putative function of *ZmHsf08* in the responses to environmental stresses, we also assessed the expression profiles of *ZmHsf08* by RT-qPCR under different stress treatments, including NaCl, PEG, and exogenous ABA. As shown in Figure 3B, the expression level of *ZmHsf08* was downregulated to different degrees after exogenous ABA treatment. Under drought stress, the *ZmHsf08* transcript level was decreased after 0 h (Figure 3C). Similarly, *ZmHsf08* expression was down-regulated following NaCl treatment (Figure 3D). These expression patterns indicated that *ZmHsf08* may function in response to abiotic stresses through the ABA-mediated pathway.

### 2.4. Overexpression of ZmHsf08 Reduces Salt and Drought Tolerance in Maize

It is suggested that *ZmHsf08* may play an important role in salt and drought stresses according to the expression pattern results. To investigate the function of *ZmHsf08* in maize, we generated *ZmHsf08*-overexpressing plants (OE8-1 and OE8-2). The transcript level of *ZmHsf08* in the transgenic lines were examined by RT-qPCR. As shown in Figure 4B, significantly increased *ZmHsf08* expression levels were observed in OE8-1 and OE8-2 plants.

To determine whether *ZmHsf08* is involved in plant salt stress tolerance, we performed a salt stress tolerance assay in transgenic lines and wild-type (WT) plants. Before stress treatments, there was no obvious difference in phenotype between the transgenic lines and WT plants. After being subjected to 200 mM NaCl treatment for 2 weeks, the most leaves of WT plants remained green, whereas the leaves of OE8-1 and OE8-2 plants were severely wilted (Figure 4A). Meanwhile, the survival rates of OE8-1 and OE8-2 plants were significantly decreased compared with the WT (Figure 4C). In addition, *ZmHsf08* expression in the transgenic lines OE8-1 and OE8-2 was downregulated under stress conditions (Appendix A). We also analyzed the physiological responses to salt stress in the WT plants and two transgenic maize lines. DAB staining showed that significantly more reddish-brown precipitate accumulated in the leaves of transgenic plants than that of WT under salt stress treatment, indicating higher H_2_O_2_ levels in transgenic lines (Figure 4D). Additionally, the contents of MDA were measured in WT and transgenic maize seedlings before and after salt stress treatment. As shown in Figure 4E, the accumulation of MDA in transgenic lines increased significantly compared with that in WT plants under both control and salt conditions. These results indicated that the overexpression of *ZmHsf08* in maize enhances sensitivity to salt stress.

To confirm the role of *ZmHsf08* in maize drought response, we performed a drought stress experiment in soil by withholding water. After drought for 2 weeks, the *ZmHsf08*-overexpressing plants exhibited more severe wilt phenotype than WT plants, and almost all their leaves were severely dehydrated and curled (Figure 5A). To examine whether the dehydration phenotype could be recovered, we further performed a re-watering experiment. After re-watering for 3 days, the WT plants had grown fresh new leaves, and about 75% of them showed growth recovery; however, only about 20% of the over-expressing plants could recover growth after re-watering (Figure 5A,B). In consistency with the phenotype, the leaves of *ZmHsf08* transgenic plants accumulated higher H_2_O_2_ levels and had higher MDA content compared to WT plants, indicating that more damage was induced in transgenic plants (Figure 5C,D). Additionally, we performed a water loss assay of detached leaves using WT and *ZmHsf08*-overexpressing plants. The rate of water loss from the transgenic plants was faster than that from the WT plants, suggesting that overexpression of *ZmHsf08* promoted water loss of leaves (Figure 5E). These findings demonstrated that overexpression of *ZmHsf08* results in decreased drought stress tolerance in maize.

### 2.5. Expression of Stress-Responsive Genes Were Altered in ZmHsf08-Overexpressing Plants

To further understand the regulation mechanisms of *ZmHsf08* in salt and drought stress responses, we assayed the expression levels of several known stress-responsive genes in WT and transgenic plants under control and stress conditions, such as *ZmDREB2A (dehydration-responsive element-binding protein 2A)*, *ZmNCED (9-cis-epoxycarotenoid dioxygenase)*, *ZmERD1 (early responsive to dehydration 1)*, *ZmRD20 (RESPONSIVE TO DESICCATION 20)*, and *ZmRAB18* (*RESPONSIVE TO ABA 18*). The *RD20* gene is often used as a stress marker gene and played an important role in plant drought tolerance [52,53]. Overexpression of *LEA (LATE EMBROGENESIS ABUNDANT PROTEIN)*, which encodes a highly hydrophilic protein, enhances plant tolerance to drought stress [54,55]. Constitutive expression of a vacuolar Na^+^/H^+^ antiporter gene, *AtNHX3 (SODIUM/HYDROGEN EXCHANGER3)*, in sugar beet (Beta vulgaris) improved high salinity tolerance in transgenic plants [56]. *NCED* and *ZEP (zeaxanthin epoxidase)* are key enzymes for ABA biosynthesis, and overexpression of the two genes enhanced abiotic tolerance in transgenic plants [57,58]. Before salt stress treatment, the expression levels of the analyzed stress-responsive genes, including *ZmDREB2A*, *ZmABF2(ABRE-BINDING FACTOR 2)*, *ZmZEP*, and *ZmNHX3*, were suppressed in transgenic plants compared with WT plants, while *ZmNCED*, *ZmERD1*, *ZmDR20*, and *ZmRAB18* expression levels in transgenic plants were higher than in WT plants. However, the expression levels of most analyzed genes (*ZmDREB2A*, *ZmERD1*, *ZmNHX3*, *ZmRAB18*, and *ZmLEA2*) were obviously decreased in transgenic plants compared with that in WT plants under salt stress (Figure 6). There was a similar pattern of repressed expression of stress-responsive genes in transgenic plants following drought stress treatment (Figure 7). The expression levels of the analyzed genes (except *ZmDREB2A*) were higher in transgenic plants compared with WT plants under control conditions to some extent; however, under drought stress, the expression levels of these genes in transgenic plants also dramatically decreased compared with those in WT plants (Figure 7). These results suggested that *ZmHsf08* plays a negative role in regulating these stress/ABA-responsive genes under salt and drought stresses.

### 2.6. Homomeric Interaction of ZmHsf08

Since *ZmHsf08* is an HSFB protein and has a core BRD sequence, we speculated that it would have no transactivation activity. To confirm this hypothesis, we performed a yeast transactivation assay. The full-length *ZmHsf08* CDS was cloned into the pGBK-T7 vector, and the recombinant construct was transformed into yeast AH109 (Figure 8A). As expected, the results showed that *ZmHsf08* has no transcriptional activity in yeast (Figure 8B).

A previous study suggested that HsfBs (HsfB1 and HsfB2b) can form homologous interactions, and are involved in the regulatory networks of stress responses through forming complexes with other proteins [47]. To determine if *ZmHsf08* forms homodimers, we performed a bimolecular fluorescence complementation (BiFC) assay. It was observed that *ZmHsf08* interacted with itself; in addition, the homodimers of *ZmHsf08* were located in the nucleus (Figure 8C). The interactions were further examined by a yeast two-hybrid (Y2H) assay. The full-length *ZmHsf08* CDS was cloned into the pGAD-T7 and pGBK-T7 vectors to generate the *ZmHsf08*-AD and *ZmHsf08*-BD proteins, respectively (Figure 8A). The *ZmHsf08*-AD was cotransformed into yeast with *ZmHsf08*-BD; as we expected, the Y2H assay also showed that *ZmHsf08* interacts with itself (Figure 8D).

## 3. Discussion

Abiotic stress, including salt and drought stress, has adverse effects on the growth and yield of crop plants. Maize, an important crop for livestock and humans, is threatened by salt and drought stress. It is known that transcription factors (TFs) play important roles in mediating abiotic stress responses by regulating the expression of stress-responsive genes [13,19]. Among these TFs, HSF TFs are an important family of regulatory proteins with diversified functions. In recent years, the functions of HSF genes involved in diverse abiotic stress responses have been reported in soybean, tomato, Arabidopsis, wheat, and rice [22,27,29,39,41,46]. In this study, we identified a class B *HSF* gene, *ZmHsf08*, from maize and analyzed its functions in response to salt and drought stress.

*ZmHsf08* is a typical HSF transcription factor with conserved domains, including a DNA binding domain (DBD), an intermediate OD (HR-A/B) region, one nuclear localization signal (NLS), and one nuclear export signal (NES). The sequence analysis revealed that *ZmHsf08* had a close homologous relationship with HsfB1 proteins from other species (Figure 1B), suggesting that *ZmHsf08* belongs to the HsfB1 subgroup. We found that *ZmHsf08* contained the NLS sequence (KKRR), which suggests a putative nuclear targeting of *ZmHsf08*, and the subcellular localization assay determined that *ZmHsf08* is a nuclear localized protein (Figure 2). In addition, *ZmHsf08* had no transcriptional activity in yeast as a result of the conserved B3 repression domain (BRD) at the C-terminus of *ZmHsf08* sequence (Figure 1B), which is consistent with the features of class B HSFs [52,53]. These results demonstrated that *ZmHsf08* is a novel *HsfB* gene in maize.

Many studies have reported that the *Hsf* genes are involved in salt and drought stress responses. For example, the Arabidopsis plants overexpressing *AtHsfA6a* and *AtHsfA7b* exhibited enhanced tolerance against salt and drought stresses [42,54]. The ectopic expression of tomato *HSFA3* in Arabidopsis increased salt hypersensitivity in transgenic plants [37]. The overexpression of *OsHsfA7* in rice demonstrated that *OsHsfA7* acted as a positive regulator in salt and drought tolerance [41]. On the contrary, *OsHsfB2b* functioned as a negative regulator in response to salt and drought stresses in rice [46]. However, the biological functions of most *HsfB* proteins in maize remains largely elusive. Interestingly, the expression profiling revealed that *ZmHsf08* expression was down-regulated after treatment with PEG, NaCl, and ABA (Figure 2B–D), suggesting that *ZmHsf08* may play a role in abiotic-stress responses in maize.

Our study demonstrated that *ZmHsf08* is involved in salt and drought stress response. Overexpression of *ZmHsf08* in maize increased sensitivity to salt and drought stress, with worse growth performance and a significantly decreased survival rate under stress treatments (Figure 4 and Figure 5). Salt and drought stress impose osmotic stress, which leads to the excess generation of reactive oxygen species (ROS), resulting in damage to plant cellular physiology and biochemistry [55,56]. H_2_O_2_ and O^−2^ are two ROS indices that are involved in abiotic stress signaling. Therefore, the accumulation of H_2_O_2_ was revealed using diaminobenzidine (DAB) staining under normal and stress conditions using maize leaves. In our study, the accumulation of H_2_O_2_ contents was obviously increased in *ZmHsf08* transgenic plants compared to WT plants under salt and drought stress conditions (Figure 4D and Figure 5C), implying that the overexpressing *ZmHsf08* in maize apparently aggravated oxidative damage by generating high ROS levels during stress treatments. This result was further confirmed by malonaldehyde (MDA) content, which is an indicator of lipid peroxidation and cell membrane damage. The MDA level in the transgenic plants was significantly higher than in WT plants in response to drought and salt stress treatments (Figure 4E and Figure 5D), indicating that *ZmHsf08* increases cell membrane injury under stress conditions. Therefore, *ZmHsf08* may play a negative role in plant abiotic stress responses.

To adapt to various abiotic stresses, especially salt and drought stresses, plants develop different response strategies, including modulation of the expression of stress-responsive genes [4,12]. In this study, we found that the expression levels of stress-responsive genes including *ZmERD1*, *ZmRD20*, *ZmRAB18*, *ZmNCED*, *ZmNHX3*, *ZmZEP*, and *ZmLEA* were higher in transgenic plants than those in WT plants under control conditions (Figure 6 and Figure 7). However, under drought stress, the expression levels of these stress-responsive genes were markedly decreased in transgenic plants compared with those in WT plants (Figure 7). Similarly, the stress-responsive genes *ZmERD1*, *ZmRD20*, *ZmRAB18*, *ZmNCED*, and *ZmLEA* also showed decreased transcript levels in transgenic plants compared with WT plants under salt conditions (Figure 6). These results indicated that *ZmHsf08* negatively regulates salt and drought stress responses, perhaps through the repressed expression of stress-responsive genes. It was reported that *HsfBs* have the capability to form homodimers in vivo and may be involved in complex regulatory networks of signaling processes and stress responses. The BiFC and Y2H assays indicated that *ZmHsf08* forms homodimer in vivo. Therefore, we speculate that the *ZmHsf08* homodimers alone or interacting with other TFs to regulate the expression of stress-responsive genes in stress responses. This hypothesis needs to be studied in depth in the future.

In conclusion, we cloned and identified a novel HsfB member, *ZmHsf08*, from maize. Overexpression of *ZmHsf08* in maize resulted in increased sensitivity to salt and drought stresses, which were associated with higher ROS levels and higher MDA contents. Furthermore, RT-qPCR analyses indicated that *ZmHsf08* negatively regulates the expression of stress/ABA-responsive genes in response to salt and drought stresses (Figure 6 and Figure 7). Our findings provide new information to better understand the function of *ZmHsf08* in plant abiotic stress responses.

## 4. Materials and Methods

### 4.1. Plant Materials and Growth Conditions

The maize inbred line B73 (seeds stored in our laboratory) was used for *ZmHsf08* gene cloning and expression analysis. The maize inbred line KN5585 (WT) and transgenic *ZmHsf08* plants were provided by Weimi Biotech Co., Ltd. and used for salt and drought stress experiments. Maize seeds were surface sterilized and germinated at 28 °C for 3 days (dark). Then, seedlings with primary root were sown in pots with soil and grown in a greenhouse (14 h/10 h of light/dark; 30 °C/25 °C of day/night) until they reached the three-leaf stage.

### 4.2. Sequence Alignment and Phylogenetic Analysis

Sequences of *ZmHsf08* and other HsfB members from different plant species (*Oryza sativa*, Sorghum bicolor, *Brachypodium distachyon*, and *Arabidopsis thaliana*) were obtained from NCBI (https://www.ncbi.nlm.nih.gov/) (accessed on 6 May 2021). The amino acid sequences of different HsfB1 proteins were aligned by ClustalX software. The conserved motifs of these proteins, including DNA binding domain (DBD), oligomerization domain (OD), nuclear export signal (NES), and nuclear location signal (NLS), were defined by SMART (http://smart.embl-heidelberg.de/) (accessed on 8 January2021).

A phylogenetic tree was constructed using the neighbor-joining method in MEGA 7.0. The neighbor-joining method was performed with 1000 bootstrap replicates.

### 4.3. Expression Analysis of ZmHsf08

The roots, stems, and leaves form seedlings of B73 at the three-leaf stage, and immature cornsilk, tassels, and bract from B73 plants in the V13 stage were sampled for tissue-specific expression analysis. For stress treatments, the B73 seedlings at the three-leaf stage were either watered with 20% (*w*/*v*) PEG6000, watered with 200 mM NaCl, or watered with 0.1 mM ABA. The leaves of seedlings under each treatment were collected at the designated time points (0, 1, 3, 6, 12, and 24 h), and the samples were immediately frozen in liquid nitrogen. Total RNA was extracted using Total RNA Extraction Reagent (Vazyme), and cDNA was synthesized using HiScript III RT SuperMix (Vazyme). RT-qPCR was performed using AceQ qPCR SYBR Green Master Mix (Vazyme) on the Thermo Scientific PikoReal 96 RT-PCR instrument. ZmGAPDH was used as internal control for maize. All the primers used for RT-qPCR are listed in Appendix A.

### 4.4. Subcellular Localization

The coding sequences (CDS) of *ZmHsf08* without or with a stop codon were cloned by PCR and then constructed into p1305-GFP and PMDC43 vectors (stored in our laboratory), respectively, for subcellular localization. The primers are shown in Appendix A. The recombined vectors *ZmHsf08*-GFP and GFP-*ZmHsf08*, or the empty vectors p1305-GFP and PMDC43, were transformed into the maize protoplasts according to the previous description [59,60]. The RFP vector, a marker for nucleus localization, was cotransformed with these constructs into maize protoplasts. After being cultured in multi-well plates for 18–24 h in the dark, the protoplasts were observed under a confocal laser scanning microscope (LSM710; Zeiss).

### 4.5. Transactivation Activity and Two-Hybrid Assays in Yeast

The full-length CDS of *ZmHsf08* was cloned into a pGAD-T7 or pGBK-T7 vector, respectively. For the transactivation activity assay, *ZmHsf08*-BD reconstructed plasmid was transformed into the yeast strain AH109. In addition, the transcriptional activation activity was examined by spot assay and X-gal staining. For the two-hybrid assays, recombined vectors were transformed in pairs in the yeast Y2HGold cells. The interaction between the two proteins was examined by spot assay. The primers are listed in Appendix A.

### 4.6. BiFC Assay

The coding sequence of *ZmHsf08* without a stop codon was cloned into pUC-SPYCE and pUC-SPYNE vectors, respectively, generating Hsf08-cYFP and Hsf08-nYFP constructs. The primers used in the assay are listed in Appendix A. Different combinations of cYFP and nYFP constructs were cotransformed with RFP into maize protoplasts. The isolation and transformation of maize protoplasts were performed according to the previous protocol [59,60]. The protoplasts were cultured for 18–24 h in the dark, and then were observed under a confocal laser scanning microscope (LSM710; Zeiss).

### 4.7. Salt and Drought Stress Experiment

For the salt stress tolerance test, the three-leaf stage maize seedlings (the transgenic lines and the WT plants) were irrigated with 200 mM NaCl solution for two weeks. Then, the phenotypes and survival rates of plants were measured.

For drought stress tolerance test, the WT and transgenic plants seedlings at three-leaf stage did not receive watering for 14 days until the plants withered. After drought for 14 days and re-watering for 3 days, the recovered conditions of the WT and transgenic lines were recorded. The survival rates of plants were measured after re-watering for 3 days under normal conditions.

### 4.8. DAB Staining and MDA Content Measurement

After salt and drought treatment for 10 days, leaves of the WT and transgenic plants were stained with diaminobenzidine (DAB) to detect the accumulation of H_2_O_2_. Leaves of the same location of maize seedlings were collected and immersed in DAB solution (1 mg/mL, pH 3.8). All the samples were incubated overnight at room temperature in darkness. The leaves were then bleached by boiling in bleach solution (ethanol: acetic acid: glycerol, 3:1:1, *v*/*v*/*v*) for 20 min to remove the chlorophyll before imaging.

The three-leaf stage seedlings of WT and transgenic plants were treated with 200 mM NaCl or watering was stopped for 10 days and then the leaves were collected. The MDA was extracted from 0.1 g maize leaf samples and measured according to the protocol from NanJing JianCheng Bioengineering Institute.

## Figures and Tables

**Figure 1 ijms-22-11922-f001:**
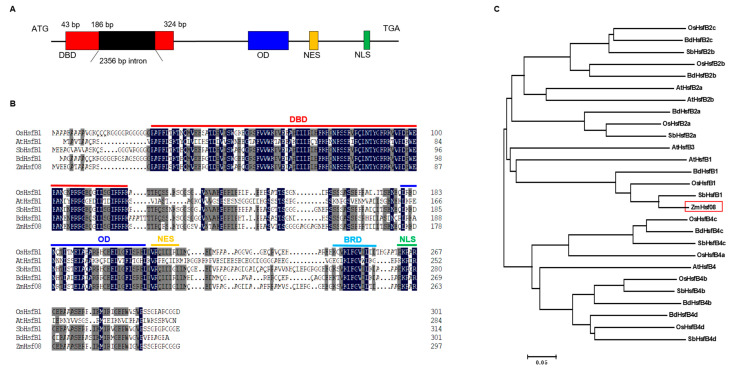
Identification of *ZmHsf08*. (**A**) Schematic representation of the *ZmHsf08* gene. A single 2356 bp intron existed in the DNA binding domain (DBD) of *ZmHsf08*. (**B**) Protein sequence alignment of *ZmHsf08* with its orthologs. (**C**) Phylogenetic analysis of *ZmHsf08* and other HsfBs proteins. The related protein names are as follows: AtHsfB1, NP_195416.1, *Arabidopsis thaliana*; OsHsfB1, XP_015611859.1, *Oryza sativa*; SbHsfB1, XP_002460330.1, *Sorghum bicolor*; and BdHsfB1, XP_003578229.1, *Brachypodium distachyon*.

**Figure 2 ijms-22-11922-f002:**
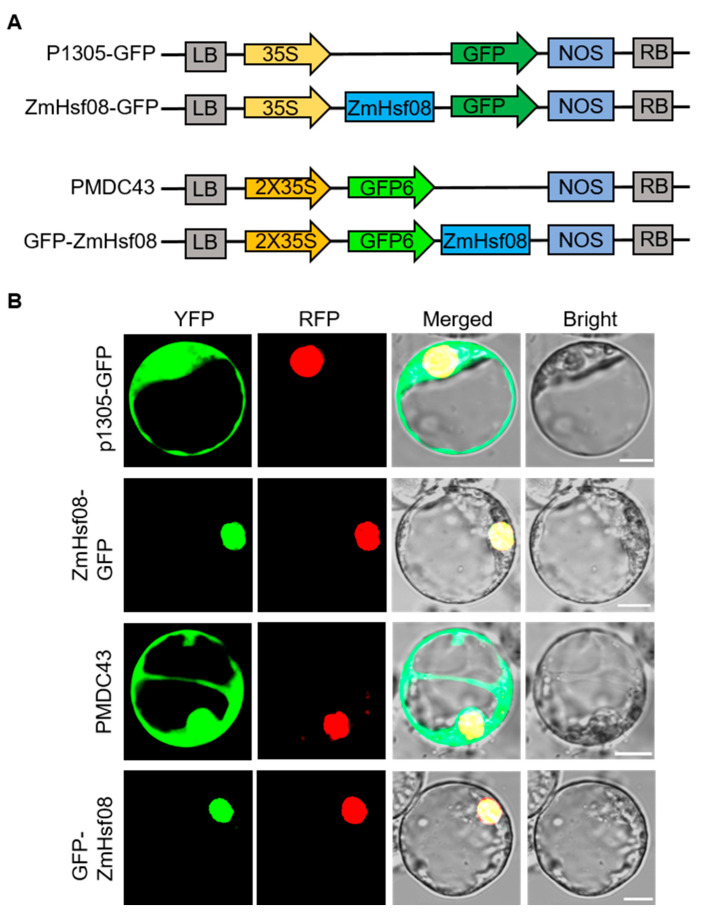
Subcellular localization of *ZmHsf08*. (**A**) Vector diagram of p1305-GFP, PMDC43, GFP-*ZmHsf08* and *ZmHsf08*-GFP. (**B**) Fusion proteins were transiently expressed in maize protoplasts. Control, p1305-GFP vector and PMDC43 vector; *ZmHsf08*-GFP and GFP-*ZmHsf08*, *ZmHsf08* fused with GFP. RFP is a marker for nucleus localization. Bars = 10 µm.

**Figure 3 ijms-22-11922-f003:**
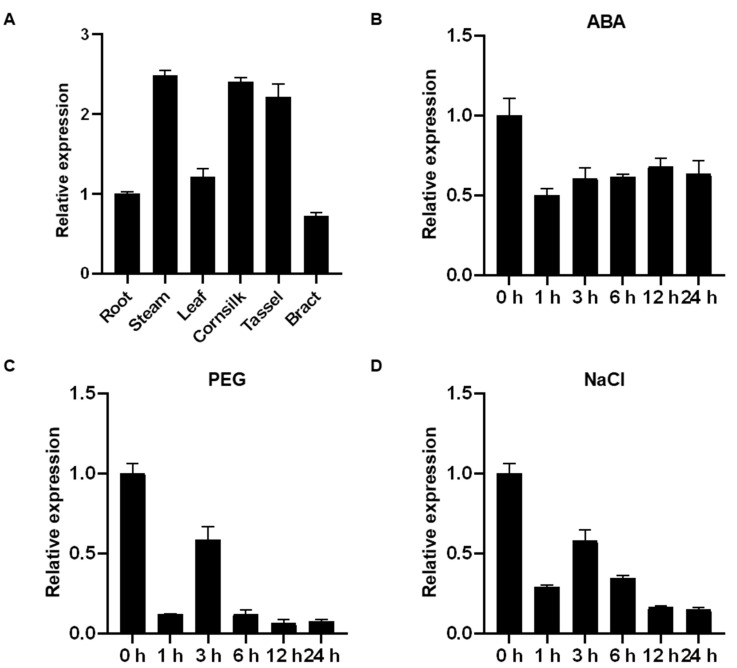
Expression patterns of *ZmHsf08*. (**A**) RT-qPCR analysis of *ZmHsf08* expression in six tissues (roots, stem, leaves, cornsilk, tassels and bract). (**B**–**D**) Expression analysis of *ZmHsf08* under ABA, PEG, and NaCl treatments. The three-leaf stage seedlings were treated with ABA, PEG, and NaCl, and the expression levels of *ZmHsf08* were detected by RT-qPCR. Bars represent means ± SD (*n* = 3 repeats).

**Figure 4 ijms-22-11922-f004:**
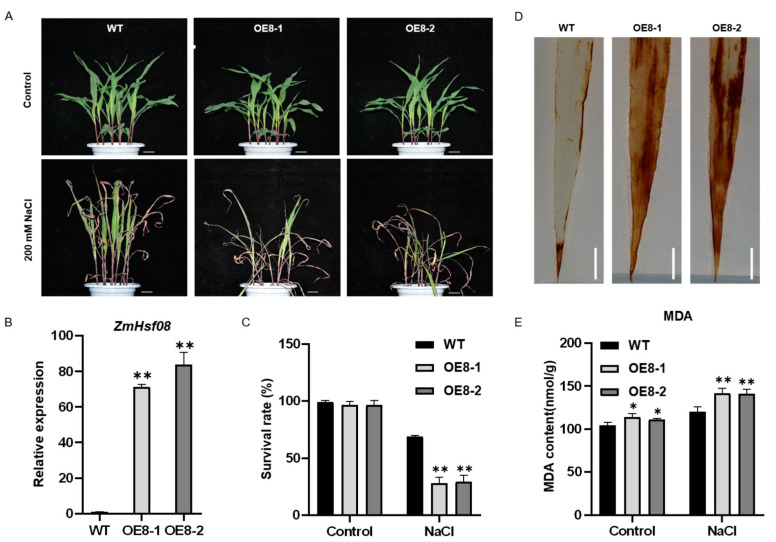
Salt stress tolerance test in the wild-type (WT) and *ZmHsf08*-overexpressing plants. (**A**) The phenotypes of WT and *ZmHsf08*-overexpressing plants under NaCl treatment. Three-leaf stage maize seedlings were treated with 200 mM NaCl for 2 weeks; the pictures were obtained before or after treatment. (**B**) Expression levels of *ZmHsf08* in transgenic maize seedlings were examined by RT-qPCR. (**C**) Survival rate of WT and *ZmHsf08* transgenic plants after treatment with 200 mM NaCl for 2 weeks. (**D**) Diaminobenzidine (DAB) staining of the maize leaves after NaCl treatment for 7 days. Bar = 1 cm. (**E**) Malondialdehyde (MDA) content of WT and *ZmHsf08* transgenic plants under normal conditions and salt treatment for 7 days. Values are means ± SD. Bars represent means ± SD (*n* = 3 repeats). Significant differences (Student’s *t* test): *, *p* < 0.05, **, *p* < 0.01.

**Figure 5 ijms-22-11922-f005:**
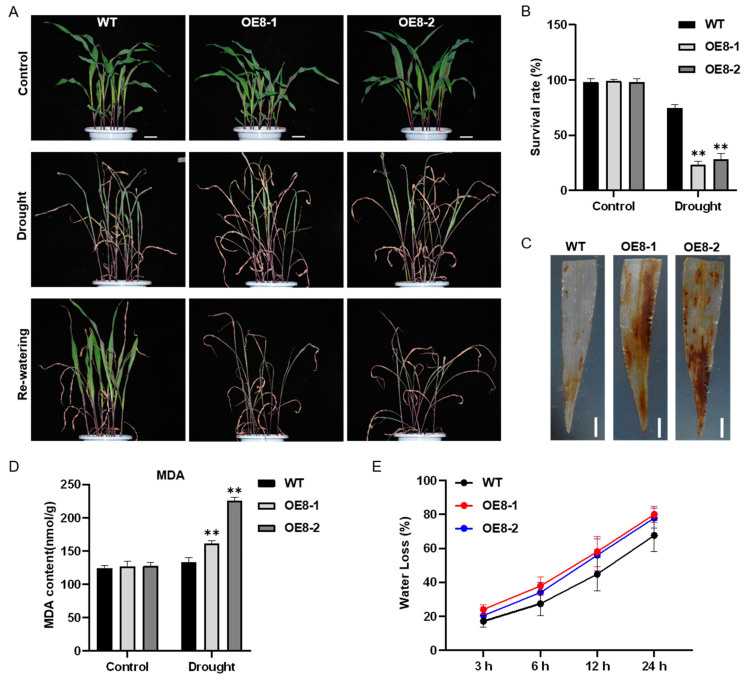
Drought stress tolerance test in the WT and *ZmHsf08*-overexpressing plants. (**A**) The phenotypes of WT and *ZmHsf08*-overexpressing plants under drought stress in soil. Three-leaf stage maize seedlings were withheld water for 2 weeks and re-watered for 3 days. The pictures were obtained under normal, drought, and re-watering conditions. (**B**) Survival rate of WT and transgenic plants after re-watering for 3 days. (**C**) DAB staining of the maize leaves. The WT and transgenic plants were withheld water for 7 days, and then the leaves were collected. Bar = 1 cm. (**D**) MDA content of WT and transgenic plants under normal condition and drought treatment for 7 days. (**E**) Water loss rate of the detached leaves of WT and transgenic plants. The leaves of three-leaf stage maize seedlings were taken, and loss of weight was measured at the indicated time point. Values are means ± SD. Bars represent means ± SD (*n* = 3 repeats). Significant differences (Student’s *t* test): **, *p* < 0.01.

**Figure 6 ijms-22-11922-f006:**
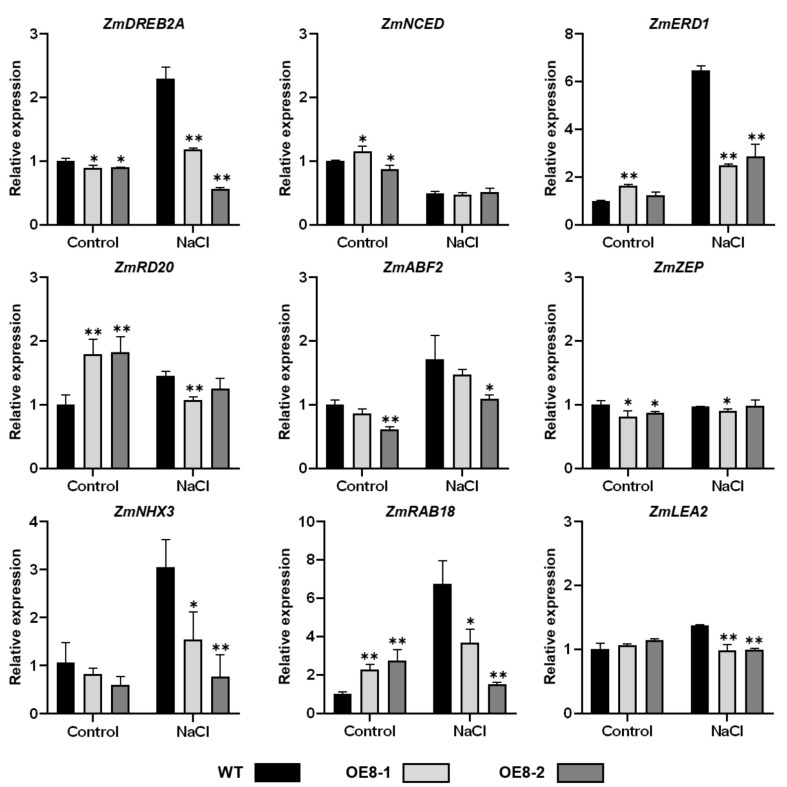
Expression patterns of stress-responsive genes in WT and *ZmHsf08*-overexpressing plants under normal condition and NaCl treatment. Three-leaf stage maize seedlings were treated with 200 mM NaCl for 7 days. The expression levels of stress-responsive genes during salt stress were analyzed by RT-qPCR. Values are means ± SD. Bars represent means ± SD (*n* = 3 repeats). Significant differences (Student’s *t* test): *, *p* < 0.05, **, *p* < 0.01.

**Figure 7 ijms-22-11922-f007:**
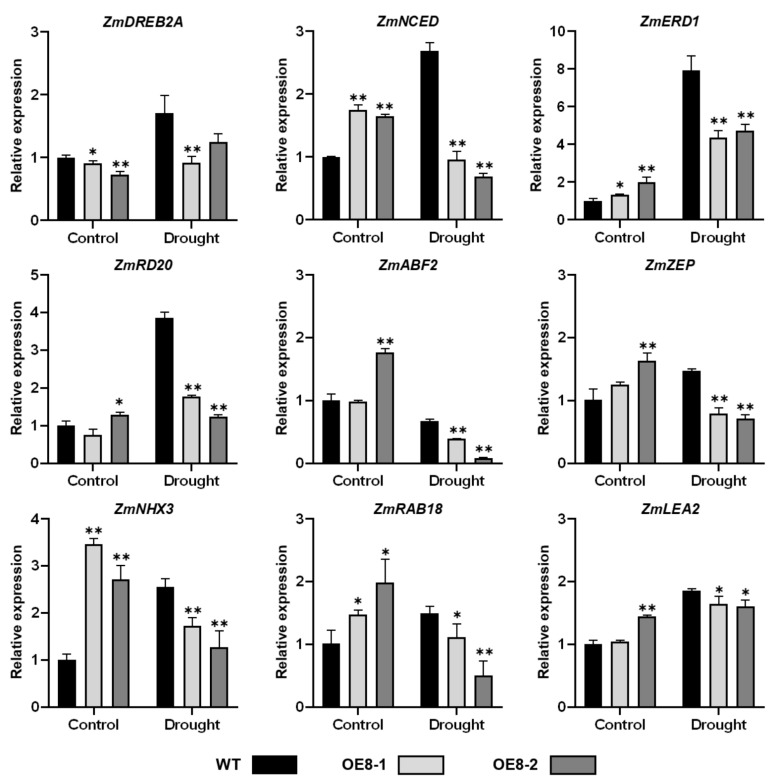
Expression levels of stress-responsive genes in WT and *ZmHsf08*-overexpressing plants under normal condition and drought treatment. Three-leaf stage maize seedlings were withheld water for 7 days. The expression levels of stress-responsive genes during drought stress were analyzed by RT-qPCR. Values are means ± SD. Bars represent means ± SD (*n* = 3 repeats). Significant differences (Student’s *t* test): *, *p* < 0.05, **, *p* < 0.01.

**Figure 8 ijms-22-11922-f008:**
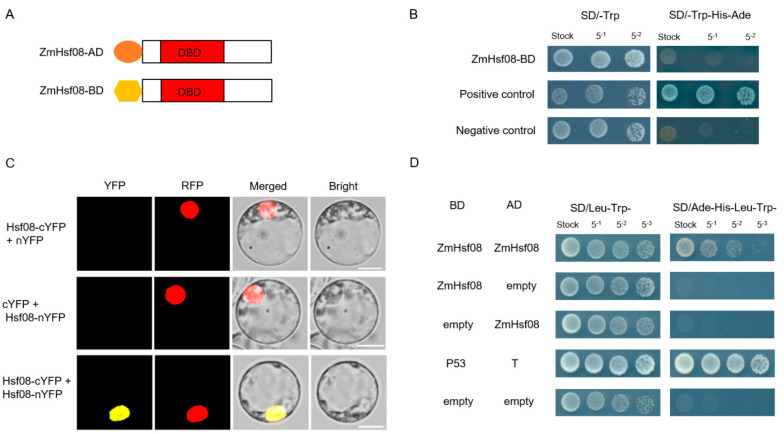
Homodimerization of *ZmHsf08*. (**A**) Schematic representation of the structures of *ZmHsf08*-AD and *ZmHsf08*-BD. (**B**) The transcriptional activation activity assay in yeast cells. (**C**) Bimolecular fluorescence complementation (BiFC) assay of the homodimerization of *ZmHsf08* in maize protoplasts. Different combinations of the nYFP and cYFP fusion constructs were cotransformed into maize protoplasts, and the fluorescence signals were examined with a confocal microscope. (**D**) Yeast two-hybrid (Y2H) analysis of the self-interaction of *ZmHsf08*. AD, GAL4 activation domain; BD, GAL4 DNA binding domain; Protein–protein interactions were examined by yeast cell growth on the SD/-Leu/-Trp/-His/-Ade plate.

## Data Availability

The data presented in this study are available on request from the corresponding author.

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
