# Peer review of "A Novel Heat Shock Transcription Factor (ZmHsf08) Negatively Regulates Salt and Drought Stress Responses in Maize"

_ijms, 2021, doi:10.3390/ijms222111922_

Round 1
Reviewer 1 Report
This study aims to study the characteristics and function of ZmHsf08. The authors performed experiments to determine the localization of ZmHsf08, sensitivity to salt and drought stress amongst others. In order to nail down the nature of ZmHsf08 and determine if it functions as transcription factor, a yeast two hybrid was performed. A BiFC assay was performed to look for homodimerization. These experiments have provided some insights into the nature and functions of ZmHsf08 but there are several unanswered questions. Several experiments have been carried out but some are inconclusive.
This paper can be made published by addressing some queries and experimentation.
Major Issues:
1. The subcellular localization of ZmHsf08 was performed and shown that ZmHsf08 localizes to the nucleus. The gene was tagged at the C-terminus without stating any reasons. Why was the gene tagged at the C-terminus? It may be possible that the GFP signal in the nucleus is not due to the expression of a full length gene. It may be only GFP or a small chunk of protein going to the nucleus.These results must be confirmed by tagging the protein at another location and carrying out a Western blot to determine the size of the protein in the nucleus, especially in the context that it does not function as a TF.
2. Figure 3: Slow decline in transcript level after ABA treatment may also be attributed to low intake of ABA after spraying on plants. Why was ABA sprayed on plants and the plants not incubated in solution as with other chemicals? For experiments in Figure 3, a uniform treatment should be given for all the treatments.
3. Figures 3 and 4: Why were the concentrations of the chemicals selected as they are? Is there a background information or supplementary data on the concentrations of chemicals selected in experiments?
4. A BiFC assay was performed to look for homodimerization although no biological significance is linked to homodimerization of the gene. BiFC is an overexpression experiment and homodimers can form.
5. It would have been interesting to see how ZmHsf108 increases the sensitivity of maize to salt and drought stress but authors want to pursue that as a future study.
Minor Issues:
1. The paper has innumerable grammar, spelling and tense errors. It appears that paper writing was rushed. It must be proof read and revised.
2. Figure 3 a: Steam should be replaced by Stem
Reviewer 2 Report
The manuscript is well structured, and the contents are easy to follow. Although it didn't significantly impact understanding, English grammar must be extensively revised. Two major issues are verb tenses, especially in the results section, and subject-verb agreement. Regarding the science, there are two questions. 1. In results section 2.4, the authors showed in figure 4B the over-expressed level of ZmHsf08 in the transgenic plants comparing to WT. But in order to prove negative impact of ZmHsf08 on salt and drought stress tolerance, it is better to show the comparative expression level of ZmHsf08 in these transgenic plants under abiotic stress. Because we learned from results section 2.3 that, under environmental stress, the expression level of ZmHsf08 was down-regulated. One may wonder what's the situation in transgenic plants after stress treatment, and if the transgenic plants still maintained a relatively higher level of ZmHsf08 than WT, to further validate the reduction effect of ZmHsf08 on salt/drought tolerance in the conclusion. 2. In results section 2.5, the authors showed the altered expression level of several stress-responsive genes in the ZmHsf08 over-expressed transgenic plants. The description of the results need to be more accurate, and not all the analyzed genes' expression level decreased in transgenic plants comparing to WT under stress treatment (line 204-206), especially in the OE8-1 strain.
Round 2
Reviewer 1 Report
Please accept the manuscript for publication
Reviewer 2 Report
Thank you for the extensive revision on English writing, which certifies this manuscript to be published in an English journal.